# Utility of Comprehensive Serum Glycopeptide Spectra Analysis (CSGSA) for the Detection of Early Stage Epithelial Ovarian Cancer

**DOI:** 10.3390/cancers12092374

**Published:** 2020-08-21

**Authors:** Koji Matsuo, Kazuhiro Tanabe, Masaru Hayashi, Masae Ikeda, Miwa Yasaka, Hiroko Machida, Masako Shida, Kenji Sato, Hiroshi Yoshida, Takeshi Hirasawa, Tadashi Imanishi, Mikio Mikami

**Affiliations:** 1Division of Gynecologic Oncology, Department of Obstetrics and Gynecology, University of Southern California, Los Angeles, CA 90033, USA; Koji.Matsuo@med.usc.edu; 2Norris Comprehensive Cancer Center, University of Southern California, Los Angeles, CA 90033, USA; 3Medical Solution Promotion Department, LSI Medience Corporation, Tokyo 1748555, Japan; tanabe.kazuhiro@mp.medience.co.jp; 4Department of Obstetrics and Gynecology, Tokai University School of Medicine, Kanagawa 2591193, Japan; hayashimasaru0620@hotmail.com (M.H.); ikedam@tokai-u.jp (M.I.); mm_4amm1302@yahoo.co.jp (M.Y.); hiroko.machida@tokai.ac.jp (H.M.); shida@is.icc.u-tokai.ac.jp (M.S.); igymichael@gmail.com (K.S.); hiroshiyoshidamd@gmail.com (H.Y.); hira@is.icc.u-tokai.ac.jp (T.H.); 5Department of Molecular Life Science, Division of Basic Medical Science and Molecular Medicine, Tokai University School of Medicine, Kanagawa 2591193, Japan; imanishi@tokai-u.jp

**Keywords:** comprehensive serum glycopeptide spectra analysis, orthogonal partial least-squares discriminant analysis, epithelial ovarian cancer, screening, cancer antigen 125, human epididymis protein 4

## Abstract

Comprehensive serum glycopeptide spectra analysis (CSGSA) evaluates >10,000 serum glycopeptides and identifies unique glycopeptide peaks and patterns via supervised orthogonal partial least-squares discriminant modeling. CSGSA was more accurate than cancer antigen 125 (CA125) or human epididymis protein 4 (HE4) for detecting early stage epithelial ovarian cancer. Combined CSGSA, CA125, and HE4 had improved diagnostic performance. Thus, CSGSA may be a useful screening tool for detecting early stage epithelial ovarian cancer.

## 1. Introduction

Ovarian cancer is the seventh most common malignancy in women worldwide, with 238,700 cases diagnosed in 2012 [1]. As women with ovarian cancer often lack specific symptoms, a large number of affected women present with advanced stage disease, wherein survival rates are dismal [2]. Hence, the early detection of ovarian cancer is an urgent unmet need in women’s healthcare.

To date, useful biomarkers for screening of ovarian cancer remain scarce [2]. In the current study, we examined the utility of comprehensive serum glycopeptide spectra analysis (CSGSA)—considering the diagnostic accuracy—for detecting early stage epithelial ovarian cancer (EOC). CSGSA evaluates >10,000 serum glycopeptides and identifies unique peaks and patterns of glycopeptides (Appendix A) via supervised orthogonal partial least-squares discriminant modeling (OPLS-DA) [3]. The results of CSGSA (OPLS-DA) modeling were compared to those of cancer antigen 125 (CA125) and human epididymis protein 4 (HE4).

## 2. Results and Discussion

First, 59 (26.2%) cases of stage I EOC were compared to 166 (73.8%) non-EOC control cases in the training set (Appendix A). According to the receiver operating characteristic curve analysis, the cutoffs were set as 25 U/mL for CA125, 53 pmol/L for HE4, and 0.8 for CSGSA (OPLS-DA). When utilizing these cutoffs, the area under the curve (AUC) for the discriminatory ability of stage I EOC over non-EOC control cases was 93% for CSGSA (OPLS-DA), higher than that for CA125 (88%) and HE4 (87%). Similar results were observed for the positive predictive value (70% for CSGSA (OPLS-DA), 55% for CA125, and 59% for HE4), sensitivity (90%, 78%, and 76%), and accuracy (87%, 78%, and 80%; Table 1).

We then applied this analytic platform (Appendix A) to the test set (Appendix A). In total, 29 (26.1%) cases of stage I EOC were compared to 82 (73.8%) non-EOC control cases (Figure 1). The AUC for distinguishing stage I EOC versus non-EOC control cases was 91% on CSGSA (OPLS-DA) modeling; the performance remained higher than that for other markers (83% for CA125 and 86% for HE4). Sensitivity was also higher on CSGSA (OPLS-DA) analysis (86%) than when other biomarkers were used (79% for both CA125 and HE4; Table 1 and Appendix A).

We examined the utility of the combination assay among these three markers in the test set (Table 1 and Appendix A and Figure 1). The combination index was calculated as (0.43 × CA125) + (0.11 × HE4) + (0.46 × CSGSA (OPLS-DA)); the cutoff was set as 0.12. The combination of all the three markers exhibited the highest AUC value (96%). Moreover, the positive predictive value for the combination of the three markers (81%) outperformed the single assay by 15–21 points (59% for CA125, 66% for HE4, and 66% for CSGSA (OPLS-DA); Table 1 and Appendix A).

CSGSA (OPLS-DA) had better accuracy than historical biomarkers (CA125 and HE4): this result is promising, highlighting the possible utility of CSGSA (OPLS-DA) as a biomarker for the detection of early stage epithelial ovarian cancer. Unlike a single marker assay such as CA125 or HE4, CSGSA (OPLS-DA) uses the pattern of a high number of glycopeptides. Although several ovarian cancer-screening tools utilize multi-marker assays [2,4], CSGSA (OPLS-DA) evaluates >1500 glycopeptides digested from serum glycoproteins. Moreover, the marker value of CSGSA (OPLS-DA) is created using OPLS-DA, which is a statistical method to separate two groups (EOC and non-EOC controls). Furthermore, when usual tumor markers are used, which are secreted by tumor cells, the biomarker amounts in serum are dependent on tumor volume. However, the result of CSGSA (OPLS-DA) does not depend on the number of tumor cells. This is a possible reason why CSGSA had a more superior performance than the other commonly used biomarkers (CA125 and HE4); hence, it is a novel method for the detection of early stage epithelial ovarian cancer.

Preoperative assessment and prediction of suspected ovarian malignancy may be useful for surgical management. In the absence of ovarian malignancy, minimally invasive surgery can be safely considered. Alternatively, in the presence of malignancy, laparotomy is recommended to decrease the risk of capsule rupture, which can negatively impact survival [5].

The limitations of the study include the small sample size and heterogeneous tumor types. The lack of external validation is another limitation, and the generalizability of this method needs to be assessed in different populations. Another limitation of this study is non-existence of clear evidence that show whether this CSGSA value (POLS-DA) could be specific for EOC or not. CSGSA (OPLS-DA) evaluates >1500 glycopeptides digested from serum glycoproteins. Moreover, the marker value of CSGSA (OPLS-DA) is created using OPLS-DA, which is a statistical method to separate two groups (EOC and non-EOC controls). If we apply serum-digested glycopeptides of other malignancies into this EOC diagnosis system of OPLS-DA, it would be meaningless because this EOC diagnosis system of OPLS-DA can work just to separate EOC and non-EOC. However, we really calculated the CSGSA value (OPLS-DA) of stage 1 cervical cancer (CC) and stage 1 endometrial cancer (EC), by which we could separate CC and EC patients from non-CC and non-EC ones (preliminary data), but not more significantly than our result between EOC and non-EOC patients. However, these data mean that CSGSA value (OPLS-DA) could differentiate cancer patients from non-cancer patients by using various target groups and control groups. For adding more organ-specific capability, we tried the combination assay with CA125 and HE4, which are EOC specific markers. We will also need to check the other organ cancers.

## 3. Materials and Methods

### 3.1. Patient Samples

A total of 88 serum samples (59 and 29 in the training and test sets, respectively) were prospectively obtained from consecutive patients with stage I EOC (Appendix A). Patients with non-EOC controls included both healthy women (*n* = 220) and patients with leiomyoma (*n* = 14) or benign ovarian tumors (*n* = 14). The inclusion criteria for the sample set of healthy women were no history of cancer and no hospitalization in the past 3 months. The study-specific exclusion criteria are shown in Appendix A. Sera was obtained by centrifuging blood samples and stored at −80 °C until CSGSA analysis to avoid repeated freeze–thaw cycles.

### 3.2. Preparation of Quality Control Serum, and Calculation of Inter- and Intra-Assay Coefficients of Variability

Detailed descriptions have been provided previously [3]. A quality control (QC) sample was prepared by pooling the sera of several women with EOC and non-EOC controls; 2 QC and 22 samples were prepared within a day, and glycopeptide expression values were obtained as the ratio between samples and the average values of two QC samples.

### 3.3. Sample Preparation for Glycoprotein Profiling

Previously described techniques were used for CSGSA [3,6,7].

### 3.4. Liquid Chromatography and Mass Spectrometry

The detailed methods for liquid chromatography and mass spectrometry have been described elsewhere [3].

### 3.5. Data Processing

Detailed descriptions regarding this issue have been reported previously [3]. Briefly, original software, “Marker Analysis,” was used to analyze all mass spectral data [8]. The peak area was defined as an area with integrating curves from beginning to end. Peak alignment was performed to maintain the error of retention time and m/z of each peak position within 0.3 min and 0.06 Da, respectively.

Calculating ratios between each peak area and the average peak areas of QCs allowed normalization of mass spectra data. Then, the mode-establishing method with SIMCA software (version 13.0.3; Umetrics; Umeå, Sweden) was applied to the normalized data [9]. The protocols developed for the software program Excel VBA were used to create heat maps of mass spectral data.

### 3.6. Pattern Recognition Analysis and Cross-Validation

Glycopeptide spectra data (Appendix A) were analyzed in a multivariate manner [9,10,11], and OPLS-DA was applied to distinguish between the EOC and non-EOC control groups (Appendix A). Before OPLS-DA, the data set was separated into training and test sets (Appendix A) to validate the training model. OPLS-DA showed two-dimensional differentiation using the first and second principal components (Figure 1 and Appendix A). OPLS-DA is a method that elicits discriminating factors between two classes, and the model is generated by reducing non-discriminable dimensions (spaces) step-by-step, thereby eliciting an underlying factor (single dimension determined in 1712-dimension space) that discriminates between two groups. We defined values of the first component as CSGSA value (CSGSA (OPLS-DA)); the values of the EOC and non-EOC control groups obtained via OPLS-DA were plotted as box-whisker plots for the training and test sets, respectively.

### 3.7. Statistical Analysis

The detailed statistical methods have been given previously [3]. *p* < 0.05 indicated statistical significance (two-tailed hypothesis). All statistical analyses were performed using Statistical Package for the Social Sciences (SPSS, version 17.0, Chicago, IL, USA) and original statistical software.

### 3.8. Study Approval

The ethics committee at Tokai University approved this study (approval number: 09R-082). Written informed consent was obtained from the patients. The data of some of the study patients were obtained from a preliminary report [3].

## 4. Conclusions

The results of this study suggest that CSGSA is more accurate than CA125 or human HE4 in detecting early stage epithelial ovarian cancer, while CSGSA, CA125, and HE4 combined exhibit improved diagnostic performance. Thus, CSGSA may be a useful screening tool for detecting early stage epithelial ovarian cancer.

## Figures and Tables

**Figure 1 cancers-12-02374-f001:**
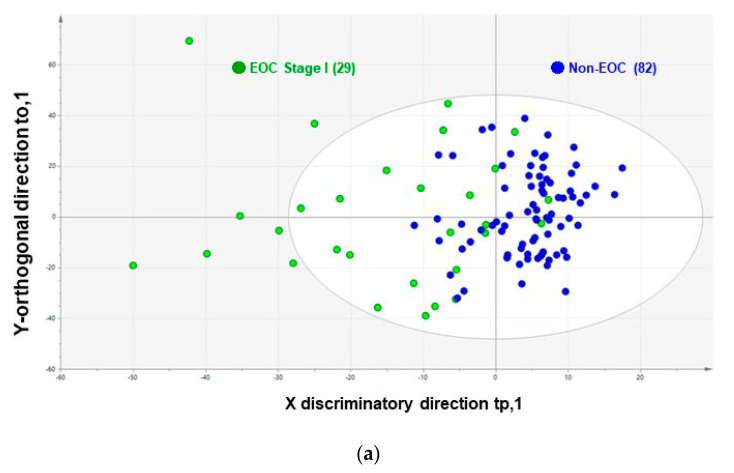
Test set: biomarker performance for the detection of stage I epithelial ovarian cancer (EOC) versus non-EOC control. (**a**) Orthogonal partial least-squares discriminant modeling (OPLS-DA) scatter plot for stage I EOC (green dots) and non-EOC control (blue dots) samples. When the test sample data set was evaluated using the trained model (Appendix A), a better differentiation was achieved in the test set between EOC (*n* = 29) and non-EOC patients (*n* = 82). Definition of comprehensive serum glycopeptide spectra analysis (CSGSA (OPLS-DA)): data structure was constructed using 1712 glycopeptide levels obtained from 225 examinees (training set: 1712 × 225 matrix), which was geometrically assumed as 225 scattered plots in the 1712-dimension space. OPLS-DA aims to find two new axes in the 1712-dimension space, which maximizes separation between the EOC and non-EOC control groups. CSGSA values (CSGSA (OPLS-DA)) were obtained via the first OPLS-DA score component, showing maximum separation of the two groups. (**b**) Box-whisker plot and receiver operating characteristic (ROC) curve of cancer antigen 125 (CA125), human epididymis protein 4 (HE4), and CSGSA (OPLS-DA) of serum samples in the test set. (**c**) ROC curve of the combination assay of cancer antigen (CA) 125, human epididymis (HE) protein 4, and CSGSA (OPLS-DA). The combination index for CA125, HE4, and CSGSA (OPLS-DA) was calculated using the following equation: combination index = (0.43 × CA125) + (0.11 × HE4) + (0.46 × CSGSA [OPLS-DA]). This combination index shows a much higher area under the curve (96%) than CA125, HE4, or CSGSA (OPLS-DA) when used alone. The cutoff of this combination index was used to maximize the sensitivity and specificity for separating stage I EOC from non-EOC controls, as defined using 0.120 obtained via ROC curve analysis in the training set. The combination index is calculated as follows: the values of CA125 and HE4 are logarithmically transformed. The transformed CA125, HE4, and CSGSA values are normalized such that the mean value is zero and the standard deviation is one. The three transformed values are then summed with weighted parameters, which are optimized using Excel powered by Solver, a sequential quadratic programming method that maximizes two-group separation under a constrained condition. The sum of the three weight parameters is one.

**Table 1 cancers-12-02374-t001:** Diagnostic performance for stage I epithelial ovarian cancer versus non-EOC controls.

Diagnostic Model	Sensitivity	Specificity	PPV	NPV	Accuracy	AUC (95% CI)
Training set						
CA125	78%	81%	55%	91%	78%	88% (83–94)
HE4	76%	80%	59%	89%	80%	87% (81–92)
CSGSA (OPLS-DA)	90%	86%	70%	96%	87%	93% (90–97)
Test set						
CA125	79%	80%	59%	92%	80%	83% (73–94)
HE4	79%	85%	66%	92%	84%	86% (78–95)
CSGSA (OPLS-DA)	86%	84%	66%	95%	85%	91% (85–98)
Combination assay						
CA125 + HE4	90%	88%	71%	95%	87%	90% (82–99)
CSGSA (OPLS-DA) + CA125	90%	91%	78%	95%	90%	95% (91–99)
CSGSA (OPLS-DA) + CA125 + HE4	90%	93%	81%	96%	92%	96% (93–100)

Abbreviations: PPV, positive predictive value; NPV, negative predictive value; AUC, area under the curve; CI, confidence interval; CSGSA, comprehensive serum glycopeptide spectra analysis; and OPLS-DA, orthogonal partial least-squares discriminant modeling.

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
