# Peer review of "Utility of Comprehensive Serum Glycopeptide Spectra Analysis (CSGSA) for the Detection of Early Stage Epithelial Ovarian Cancer"

_cancers, 2020, doi:10.3390/cancers12092374_

Round 1
Reviewer 1 Report
In this MS, the authors developed CSGSA and evaluated it as a biomarker of early EOC.
The authors should address below before the publication
#1
The big problem is the exclusion of the other malignancy. Please add the comparison data of CSDSA in EOC to that in other malignancies. If other malignancy have the same tendency to this CSGSA, the lack of specificity to EOC is the problem.
#2
Women will often use hormonal agent. Why do you exclude the history of hormonal agent. Do you have any data that CSGSA did not work in that population???
If you exclude the patients with the history of hormonal agent, this method didi not use in the clinic.
Author Response
Thank you for giving us questions and comments.
Our answers are as follows,
#1
The big problem is the exclusion of the other malignancy. Please add the comparison data of CSGSA in EOC to that in other malignancies. If other malignancy have the same tendency to this CSGSA, the lack of specificity to EOC is the problem.
Reply:
I added the below sentence in the results and discussion section.
The another limitation of this study is non existence of clear evidence that show this CSGSA value (POLS-DA) could be specific for EOC or not. CSGSA (OPLS-DA) evaluates >1500 glycopeptides digested from serum glycoproteins. Moreover, the marker value of CSGSA (OPLS-DA) is created using OPLS-DA, which is a statistical method to separate two groups (EOC and non-EOC controls). If we apply serum digested glycopeptides of other malignancies into this EOC diagnosis system of OPLS-DA, it would be meaningless because this EOC diagnosis system of OPLS-DA can work just to separate EOC and non EOC. However, we really calculated the CSGSA value (OPLS-DA) of stage1 cervical cancer (CC) and stage1 endometrial cancer (EC), which turned to separate CC and EC patients form non CC and Non EC ones (preliminary data), but not more significantly than our result between EOC and non EOC patients. However, these data means that CSGSA value (OPLS-DA) could be differentiate cancer patient from non-cancer patients by using various target groups and control groups. For adding more organ-specific capability, we tried the combination assay with CA125 and HE4, which are EOC specific markers. We will also need to check the other organ cancers.
#2
Women will often use hormonal agent. Why do you exclude the history of hormonal agent. Do you have any data that CSGSA did not work in that population??? If you exclude the patients with the history of hormonal agent, this method did not use in the clinic
Reply:
We do not have data you mentioned. But there are no difference between the data of our healthy women from premenopausal and postmenopausal status. So our results are just the first step of CSGSA (OPLS-DA). We hope that we will extend the sample size and categorized the women who have the history of hormonal agent or not in future.

Reviewer 2 Report
Genaral Comments: This research is very interesting. I want to know the reason why the aurhors
submmited this paper as " Brief Report."
Specific Comments:
1) On line 43-48, the authors write "When utilizing these cutoffs, the area under the curve (AUC) for the discriminatory 44 ability of stage I EOC over non-EOC control cases was 93% for CSGSA (OPLS-DA), higher than that for CA125 (88%) and
HE4 (87%). Similar results were observed for the positive predictive value (70% for CSGSA [OPLS47-DA], 55% for CA125, and 59% for HE4); sensitivity (90%, 78%, and 76%); and accuracy (87%, 78%, and 80%; Table 1).
On line 54- 58, The AUC for distinguishing stage I EOC versus non-EOC control cases was 91% on CSGSA (OPLS-DA) modeling; the performance remained higher than that for other markers (83% for CA125 and 86% for HE4).
Sensitivity was also higher on CSGSA (OPLS-DA) analysis (86%) than when other biomarkers were used (79% for both CA125 and HE4; Table 1 and Table S2).
On line 88- 90, Moreover, the positive predictive value for the combination of the three markers (81%) outperformed the single assay by 15-21 points (59% for CA125, 66% for HE4, and 66% for CSGSA [OPLS-DA]; Table 1 and Table S2).
Are these data significantly different? Please show p-value in each data.
2) In materials and methods, the authors should describe the histological subtypes of 88 ovarian cancers at stage 1a.
Author Response
Thank you for giving us questions and comments.
Our answers are as follows,
Specific Comments:
1) On line 43-48, the authors write "When utilizing these cutoffs, the area under the curve (AUC) for the discriminatory ability of stage I EOC over non-EOC control cases was 93% for CSGSA (OPLS-DA), higher than that for CA125 (88%) and
HE4 (87%). Similar results were observed for the positive predictive value (70% for CSGSA [OPLS47-DA], 55% for CA125, and 59% for HE4); sensitivity (90%, 78%, and 76%); and accuracy (87%, 78%, and 80%; Table 1).
On line 54- 58, The AUC for distinguishing stage I EOC versus non-EOC control cases was 91% on CSGSA (OPLS-DA) modeling; the performance remained higher than that for other markers (83% for CA125 and 86% for HE4).
Sensitivity was also higher on CSGSA (OPLS-DA) analysis (86%) than when other biomarkers were used (79% for both CA125 and HE4; Table 1 and Table S2).
On line 88- 90, Moreover, the positive predictive value for the combination of the three markers (81%) outperformed the single assay by 15-21 points (59% for CA125, 66% for HE4, and 66% for CSGSA [OPLS-DA]; Table 1 and Table S2).
Are these data significantly different? Please show p-value in each data.
Reply:
I added the p-values in Figure S1B, Figure 1B and 1C. I include these revised PPT slides.
2) In materials and methods, the authors should describe the histological subtypes of 88 ovarian cancers at stage 1a.
Reply:
I showd the histological subtypes of 88 ovarian cancers I Table S1 in Supplementary materials

Reviewer 3 Report
Matsuo K and colleagues propose a new method for early diagnosis of ovarian cancer. Early diagnosis could be the key to prevent the high mortality rate of this type of gynecological cancer and, therefore, this type of studies are very well welcome!
The study of the serum glycopeptide profile is EOC patients versus non-EOC control revealed that a algorithm combination of CA125, HE4 and CSGSA that detects 96% of the EOC at stage I in serum samples. However, it's not clear in this brief report, whether CSGSA individual results are specific for EOC or are elevated in cancer patients in general?
Another issue that mus be addressed as a limitation is the fact that in the EOC group the majority of the cases are endometrioid and clear cell tumours (N=28 and n=38, respectively). Serous EOC is the most lethal type of OC, in most of the cases detected in advanced cases and perhaps this was a reason for the low number of serous OC at stage I included in this study. This algorithm of detection should be evaluated in the different types of EOC separately.
Author Response
Thank you for giving us questions and comments.
Our answers are as follows,
〇The study of the serum glycopeptide profile is EOC patients versus non-EOC control revealed that a algorithm combination of CA125, HE4 and CSGSA that detects 96% of the EOC at stage I in serum samples. However, it's not clear in this brief report, whether CSGSA individual results are specific for EOC or are elevated in cancer patients in general?
Reply:
I added the below sentence in the results and discussion section.
The another limitation of this study is non existence of clear evidence that show this CSGSA value (POLS-DA) could be specific for EOC or not. CSGSA (OPLS-DA) evaluates >1500 glycopeptides digested from serum glycoproteins. Moreover, the marker value of CSGSA (OPLS-DA) is created using OPLS-DA, which is a statistical method to separate two groups (EOC and non-EOC controls). If we apply serum digested glycopeptides of other malignancies into this EOC diagnosis system of OPLS-DA, it would be meaningless because this EOC diagnosis system of OPLS-DA can work just to separate EOC and non EOC. However, we really calculated the CSGSA value (OPLS-DA) of stage1 cervical cancer (CC) and stage1 endometrial cancer (EC), which turned to separate CC and EC patients form non CC and Non EC ones (preliminary data), but not more significantly than our result between EOC and non EOC patients. However, these data mean that CSGSA value (OPLS-DA) could be differentiate cancer patient from non-cancer patients by using various target groups and control groups. For adding more organ-specific capability, we tried the combination assay with CA125 and HE4, which are EOC specific markers. We will also need to check the other organ cancers.
〇Another issue that must be addressed as a limitation is the fact that in the EOC group the majority of the cases are endometrioid and clear cell tumours (N=28 and n=38, respectively). Serous EOC is the most lethal type of OC, in most of the cases detected in advanced cases and perhaps this was a reason for the low number of serous OC at stage I included in this study. This algorithm of detection should be evaluated in the different types of EOC separately.
Reply:
Thank your for giving me the great suggestion to us. However, majority of the histological type of EOC in Japan is clear call and endometrial tumors. Currently, we are starting to collect EOC serum samples nationwide. So, in the next step, Early detection dependent on histological types of EOC will be our target.

Round 2
Reviewer 1 Report
Now the authors address the requirements.